# Finite Element Solution for Dynamic Mechanical Parameter Influence on Underwater Sound Absorption of Polyurethane-Based Composite

**DOI:** 10.3390/ijms232314760

**Published:** 2022-11-25

**Authors:** Dexian Yin, Yue Liu, Yimin Wang, Yangyang Gao, Shikai Hu, Li Liu, Xiuying Zhao

**Affiliations:** 1Key Laboratory of Beijing City on Preparation and Processing of Novel Polymer Materials, Beijing University of Chemical Technology, Beijing 100029, China; 2Beijing Engineering Research Center of Advanced Elastomers, School of Materials Science and Engineering, Beijing University of Chemical Technology, Beijing 100029, China

**Keywords:** dynamic mechanical parameter, underwater sound absorption, finite element, polyurethane

## Abstract

Underwater noise pollution, mainly emitted by shipping and ocean infrastructure development of human activities, has produced severe environmental impacts on marine species and seabed habitats. In recent years, a polyurethane-based (PU-based) composite with excellent damping performance has been increasingly utilized as underwater sound absorption material by attaching it to equipment surfaces. As one of the key parameters of damping materials, dynamic mechanical parameters are of vital importance to evaluating the viscoelastic damping property and thus influencing the sound absorption performance. Nevertheless, lots of researchers have not checked thoroughly the relationship and the mechanism of the material dynamic mechanical parameters and its sound absorption performance. In this work, a finite element model was fabricated and verified effectively using acoustic pulse tube tests to investigate the aforementioned issues. The influence of the dynamic mechanical parameters on underwater sound absorption performance was systematically studied with the frequency domain to reveal the mechanism and the relationship between damping properties and the sound absorption of the PU-based composite. The results indicate that the internal friction of the molecular segments and the structure stiffness were the two main contributors of the PU-based composite’s consumption of sound energy, and the sound absorption peak and the sound absorption coefficient could be clearly changed by adjusting the dynamic mechanical parameters of the composite. This study will provide helpful guidance to develop the fabrication and engineering applications of the PU-based composite with outstanding underwater sound absorption performance.

## 1. Introduction

Noise pollution, considered as one of the four primary kinds of environmental pollution issues besides solid waste, water, and air pollution, not only endangers human health but also destroys the ecological system [1,2,3]. Notably, underwater noise pollution has increasingly become one of the most severe ecological problems and has received extensive attention around the globe over the past several decades, owing to the expansion of modern maritime industries and sea transportation [4,5,6]. Specifically speaking, underwater noise radiated by large ships induces physiological variation in marine organisms, which could even lead to their tissue damage and death [7]. Particularly, underwater noise pollution influences the capacity of aquatic animals to hear and employ sound, resulting in the disturbance of survival behavior [8]. Underwater noise profoundly affects the communication, hunting, and navigation of cetaceans, which depend on acoustic cues for their survival in the marine environment [9]. Furthermore, the stealth performance of warships has become an important direction in military fields, in which the key to realizing sound stealth technology is restraining the hull vibration and removing the underwater noise [10]. Consequently, eliminating noise pollution and improving the marine environment are essential whether it is in civilian or military fields.

Over recent decades, sound absorption layers attached to ships have been gradually utilized to reduce the transmission of underwater sound, which could availably increase the sound wave absorption and thus isolate the noise inside ships as well as suppress the ship vibration [11,12,13,14]. Currently, damping materials are most commonly utilized as sound absorption layers to reduce vibration and sound noise, which include viscoelastic materials, composites, metal, and intelligent damping materials [15]. Among them, viscoelastic materials display the most typical damping materials and possess excellent damping performances [16,17,18,19,20] because, compared to other damping materials, their characteristic impedance is close to that of water, so they can convert vibration and sound energy to heat energy significantly [21,22]. The PU-based composite, as a type of viscoelastic damping material, owns a particular long-chain molecular structure, which can be used as an underwater sound absorption material [23,24,25]. The clearest advantage of the PU-based composite is that its sound absorption performance can be well-regulated via molecular designing, including choosing a variety of molecular chain segments and chemical synthesis methods, adjusting the molecular chain proportion of the hard and soft segments, and incorporating them with other polymers [20,26]. Thus, the PU-based composite exhibits a promising outlook for application in sound absorption fields.

As is well-known, the dynamic mechanical parameters are of vital importance to evaluate the viscoelastic performance and influence the sound absorption properties of materials [13,27,28]. Subject to restrictions on experimental conditions and materials, it is difficult to prepare multiple viscoelastic materials with a stepped difference in dynamic mechanical parameters; consequently, it can be a challenge to fabricate sound absorption layers with controllable and devisable sound absorption performances [29,30]. Investigating the underwater sound absorption property of materials via theoretical analysis and numerical calculation is currently a fresh approach to addressing the issue [11,31]. Nevertheless, although many researchers had surveyed underwater sound absorption layers using the aforementioned methods, the systematic study on the effect of material viscoelastic parameters on the sound absorption performance of underwater layers is still seldom. Furthermore, in the modeling of the underwater sound absorption performance of layers, researchers often assumed that the dynamic modulus is a real quantity and ignored its imaginary part [32,33], and thus were likely to cause erroneous results during the calculation process. To forecast the underwater sound absorption performances of layers, the actual dynamic modulus parameters are necessary for the process of modeling and computation.

In this work, a finite element model of the underwater sound absorption layer was presented to calculate the acoustic performances of polyurethane-based (PU-based) viscoelastic materials. Owing to the advantages of providing cross-linking points for reactions with divers’ vulcanizing agents, a type of new millable polyurethane was successfully prepared, in which raw polyurethane could be vulcanized with a variety of molds. Subsequently, the sound absorption performance was experimentally measured via an acoustic pulse tube, and the results were compared with the calculation results to testify the validity of the designed finite element model. Subsequently, different gradient dynamic mechanical parameters were assumed to systemically investigate the underwater sound absorption proprieties and the sound absorption mechanisms of the composite in different frequency bands. This work could offer potential guidance for the design and application of underwater sound absorption layers.

## 2. Results and Discussion

### 2.1. Effect of E′ on Sound Absorption Performances

In Figure 1a, the E0 curve was the E′ of the prepared PU-based composite (the same as the E′ curve in Figure 11), and E1 to E4 were hypothetic E′ with different gradients. Figure 1b exhibits the influence of E′ on the SAC of the PU-based composite. As shown in Figure 1b, all the curves with different E′ exhibited one or two absorption peaks with the increase in frequency, which were the resonance absorption peaks of sound waves. The E′ affected the absorption curve from the initial frequency. The first and second absorption peaks of the E4 curve were at the frequency of about 1.5 and 5.5 kHz. With the E′ of the PU-based composite increased from E4 to E1, the first absorption peak frequency gradually shifted from about 2 to 8 kHz, and the second absorption peak shifted rightward until it disappeared in the studied frequency. The reason was that the increased E′ enhanced the integral stiffness of the composite, leading to an increase in the resonance frequency [34]. Meanwhile, the increased E′ improved the peak intensity of the resonance absorption peaks, indicating that a relatively high E′ could bring about an improved sound absorption performance (especially at higher frequencies) [35]. For instance, the SAC of the first absorption peak in Figure 1b increased from 0.81 (in the E4 curve) to 0.98 (in the E1 curve). Additionally, the results demonstrate that the PU-based composite with low E′ could be utilized at a relatively low frequency to realize the available absorption of sound waves.

To further investigate the energy loss mechanism of the PU-based composite at different frequencies, the effects of different E′ on the total power dissipation density at 1.5 and 10 kHz were calculated and shown in Figure 2 and Figure 3, respectively.

It was found in Figure 2 that all of the pictures displayed similar results, i.e., the distribution of the total power dissipation density was mainly lost at the top of the composite, as there was a considerable proportion of the energy loss when the sound wave was normal incident upon the surface of the composite from water [22]. Therefore, there was little energy remaining when the sound wave propagated to the bottom of the composite. Moreover, Figure 2 exhibits that the value of the total power dissipation density rose regularly and substantially when the E′ of the composite reduced from E1 to E4, in which the total power dissipation density of the E3 and E4 increased by an order of magnitude compared to E1, E2, and E0. Combined with the SAC results at low frequencies (Figure 1b), the phenomenon illustrated that the internal friction of the molecular chain segments was the main reason for the outstanding sound absorption performances of the composite with relatively low E′ [36]. At low frequencies, the softer the composite was, the greater the internal friction (generated by the viscous resistance of the viscoelastic material) was; thus, more internal energy was generated to absorb the sound energy. In contrast, the part of the composite that dissipated sound energy was reduced at the frequency of 10 kHz with the gradual reduction in E′ from E1 to E4, as presented in Figure 3. In addition, the total power dissipation density value of the composite at 10 kHz showed no obvious change with the decrease in E′, but its sound absorption property decreased (Figure 1b). The results indicate that at high frequencies, most of the sound energy could not be consumed by the internal friction of the molecular segments, whereas the resonance produced by the enhanced structure stiffness (i.e., enhanced E′) was the major contributor to the consumption of sound energy.

### 2.2. Effect of tanδ on Sound Absorption Performances

The tanδ is defined as the ratio of the imaginary part to the real part of the dynamic modulus, in which the imaginary part represents the loss modulus of materials, and the real part is known as the E′ of materials. In Figure 4a, the tanδ0 curve was the same as the tanδ curve in Figure 11, and tanδ1 to tanδ4 were hypothetic tanδ with different gradients. The effect of tanδ on the SAC was investigated and is displayed in Figure 4b.

In Figure 4b, it was observed that all of the curves displayed the same trends, i.e., the curves first increased and then slightly decreased with the increasing frequency from 0 to 10 kHz. The variation in tanδ showed little influence on the frequency of the sound absorption peaks, which were presented at about 4 kHz. To survey the sound absorption mechanism of the PU-based composite at the frequency of the sound absorption peak, the displacement vector and the displacement field of the composite with different tanδ at 4 kHz were investigated and are exhibited in Figure 5. It was obvious that the displacement direction (red arrow in Figure 5) and displacement size showed little change with different tanδ, illustrating that the change in tanδ could not affect the overall vibration characteristics of the composite [37]. It was probably because the position of the sound absorption peak was more related to the material structure (geometric size); however, the change in tanδ only reflected the change in the material parameter (tanδ), and its structure did not change [38].

Furthermore, Figure 4b shows that the SAC of the sound absorption valley (at the frequency of about 7 kHz) improved gradually with the increased tanδ, and the curve of tanδ1 possessed the highest SAC compared with the curves of other tanδ. The total power dissipation density at 7 kHz is exhibited in Figure 6. As shown in Figure 6, the total power dissipation density presented the same trends as Figure 4b, i.e., the biggest tanδ meant the highest power consumption. In addition, the part of the composite that dissipated sound energy was gradually decreased at 7 kHz with the reduction in tanδ from tanδ1 to tanδ4. The results demonstrate that although the increase in tanδ could not affect the overall vibration characteristics of the composite, the stress–strain hysteresis effect became obvious; thus, the energy dissipation capacity of the composite was enhanced [23,34]. In other words, more sound wave energy was converted to internal energy and simultaneously dissipated to overcome internal resistance at the same frequency, resulting in improving sound absorption proprieties.

### 2.3. Effect of ρ on Sound Absorption Performances

Density (ρ) is one of the basic properties of composites which influences the damping and sound absorption performances. The SAC results of the PU-based composite with different ρ are shown in Figure 7. When ρ of the composite increased from 950 to 1250 kg/m^3^, it was found that the position of the first sound absorption peak slightly moved to a low frequency, and the peak value showed no significant change, showing that the change in ρ had little effect on the sound absorption performance of the composite at low and intermediate frequency ranges. Additionally, the SAC improved gradually with the increase in ρ in the frequency range from 7 to 10 kHz, meaning that the increase in density could enhance the sound absorption performance of the composite at high frequencies. In general, the results demonstrate that ρ has less effect on the sound absorption performance of the PU-based composite compared with E′ and tanδ.

## 3. Methods and Materials

### 3.1. Theory and Model

#### 3.1.1. Theoretical Analysis

The outer space of the sound absorption layer was assumed as a water domain with semi-infinite space, possessing the ideal fluid condition of uniform, incompressible, nonviscous, and zero-flow velocity. The acoustic excitation was a small disturbance near the stable state of the fluid domain, and the wave equation of the fluid medium could be expressed as [38,39]:$$\frac{1}{c^{2}}\frac{\partial^{2}p}{\partial t^{2}}-\nabla^{T}\left(\nabla p\right)=0$$
in which *c* represented the sound velocity of the fluid medium ($c=\sqrt{K/\rho_{0}}$, K was the bulk modulus, and ρ_0_ was the density), *p* denoted the sound pressure, and $\nabla^{T}$ was the transpose of the gradient operator. Taking the integral of small incremental sound pressure over the fluid domain, the following equation could be obtained:$${∭}_{V_{f}}\frac{1}{c^{2}}\delta p\frac{\partial^{2}p}{\partial t^{2}}dV+{∭}_{V_{f}}\left(\nabla^{T}\delta p\right)\left(\nabla p\right)dV=\iint_{S_{sf}+S_{ff}}{\left[n\right]}^{T}\delta p\left(\nabla p\right)dS$$where *V*_f_ was the fluid domain, *S*_sf_ was the interface of the fluid domain and the solid domain, *S*_ff_ was the interface of the fluid domain and the fluid domain, and [n] was the outward unit normal vector of the surface. The equation utilized to represent the medium movement in the small amplitude linear acoustic fields was expressed as [39]:$$\rho_{0}\frac{\partial u}{\partial t}=-\nabla p$$
in which *u* represented the particle vibration velocity of the fluid medium. As the structure of the interior sound absorption layer displayed a periodic array, the sound absorption proprieties of the layer could be analyzed using a structural unit according to the Bloch’s theory. The periodic boundary term of units was set to Floquet, and only one single unit needed to be considered in the calculation. The Floquet boundary term was expressed as follows:$${\vec{u}}_{d}=\exp\left[-i{\vec{k}}_{F}\left({\vec{r}}_{d}-{\vec{r}}_{s}\right)\right]{\vec{u}}_{s}$$
where ${\vec{r}}_{d}$ and ${\vec{r}}_{s}$ were the position of the intermediate field, ${\vec{u}}_{d}$ and ${\vec{u}}_{s}$ were the dependent variable vectors, the subscripts *d* and *s* represented the target boundary and the source boundary, and ${\vec{k}}_{F}$ represented the periodicity of excitation space. Finally, the finite element equation of fluid could be written as [39]:$$\left[M^{p}\right][\ddot{p]}+[K^{P}]\left[p\right]+\rho_{0}\left[R\right][\ddot{\delta ]}=\left[\Phi \right]$$
in which $\left[M^{p}\right]$ was the mass matrix, $\left[M^{p}\right]$ was the overall stiffness matrix, $\rho^{0}$ was the density of the fluid medium, [*R*] was the integrated overall coupling matrix, p represented the sound pressure, and [δ] was the solid particle displacement. [Φ] denoted the effect of the sound field in the bounded fluid domain on the fluid interface because of the existence of the sound field in the outer unbounded fluid domain.

Sorting the corresponding fluid nodes in the elements $\left[M^{p}\right]$, $\left[K^{p}\right]$, and [R], consistent with the order of the fluid nodes corresponding to the elements in [Φ], the final discrete equation was expressed as follows [10,40]:$$\left[\begin{array}{cc} -{\left[R\right]}^{T} & [K^{s}]-\omega^{2}[M^{s}]\\ \left[K^{p}]-[C_{\emptyset }]-\omega^{2}[M^{p}\right] & -\rho_{0}\omega^{2}\left[R\right] \end{array}\right]\left\{\begin{array}{c} \left[p\right]\\ \left[\delta \right] \end{array}\right\}=\left\{\begin{array}{c} \left[F^{S}\right]\\ \left[C_{0}\right] \end{array}\right\}$$
in which $\left[M^{s}\right]$ and $\left[K^{s}\right]$ were the solid mass matrix and stiffness matrix, and $\left[C_{0}\right]$ was the boundary equivalent nodal load matrix. When the plane sound waves incident perpendicularly on the sound absorption layers, the sound reflection coefficient (*R*) could be calculated using the following equations:$$R=\frac{Z_{b}-Z_{0}}{Z_{b}+Z_{0}}$$
$$Z_{0}=\rho_{0}c_{0}$$
$$Z_{b}=\frac{p}{v}$$
in which $Z_{0}$ and $c_{0}$ displayed the acoustic impedance and sound speed in the medium, $Z_{b}$ represented the mechanical impedance per unit area of the layer surface, and v was the normal velocity of the node on the solid side. In this work, the sound absorption layer was coupled to air, and the impedance difference was large; thus, the transmission coefficient could be negligible. As a result, the sound absorption coefficient (SAC, α) could be written as [11,33]:$$\alpha =1-{\left|R\right|}^{2}$$

#### 3.1.2. Structure Design and Modeling

A unit cell of the underwater sound absorption structure is established in Figure 8, and an illustration of the structure is drawn simultaneously to describe the model for better understanding. The structure of the sound absorption layer was made up of the PU-based composite, and the medium on one side of the composite was the water phase, whereas the medium on the other side was the air phase. In particular, the water and air phases were modeled via the pressure acoustic module, whereas the sound absorption material was modeled by the solid mechanics module. To simulate the boundary interactions between the solid and fluid domains, sound structural boundary conditions were utilized in this model. The lateral size of the entire structure was taken as spatially infinite, and the sides of the structure were set as symmetry constraints to simulate that the harmonic plane waves were vertical incidence from the infinite water domain to the infinite solid domain. In addition, the nonreflecting boundary condition was utilized at the source and terminal surfaces, ensuring that only once incidence and reflection waves existed in the transmission process. The corresponding material and geometric parameters are listed in Table 1.

### 3.2. Experimental Validity of the Model

#### 3.2.1. Composite Preparation and Dynamic Mechanical Measurements

To verify the accuracy of the sound calculation results, the PU-based composite was fabricated, and the relevant viscoelastic parameters were tested. The polyurethane was prepared as follows. First, methyl tetrahydrofuran (MPTMG) (Xiaoxing Chemical Jiaxing Co., Ltd., Jiaxing, China), as the soft segment, was dehydrated for 1.5 h at 115 °C in a vacuum environment. Then, 4, 4-diphenylmethane diisocyanate (Acros, US), as the hard segment, was incorporated with MPTMG at 65 °C for 5 min to complete the prepolymerization process. The prepolymer was subsequently mixed with unsaturated chain extender 1,1,1-trimethylolpropane monoallyl ether (Acros, US) at 115 °C for 1 h, and then the post-cure reaction was conducted at 100 °C for 12 h. Finally, the internal mixer was utilized to mix raw rubber, sulfur, and other vulcanizing accelerators, and the PU-based composite was obtained after a vulcanization process of 20 min at 150 °C. The synthetic route of the PU-based composite is displayed in Figure 9.

Fourier transform infrared (FTIR) spectra of the PU-based composite were tested in attenuated total reflection mode on an FTIR spectrophotometer with a resolution of 4 cm^−1^ (Tensor 27, Bruker, Germany). Figure 10a shows the appearance of the absorption peak of the imino (N-H) (3300 cm^−1^), hydrocarbon (2900 cm^−1^), and ether groups (-O-) (1100 cm^−1^). Additionally, the absorption peak of isocyanate group (-NCO) (2300 cm^−1^) disappeared. Combined with the result of GPC curve in Figure 10b, it was clear that the PU-based composite successfully synthesized. Dynamic mechanical analyzer (DMA) (Q800, TA Instruments, USA) was used to measure the dynamic mechanical properties of the PU-based composite in the range of −100–100 °C at a rate of 3 °C/min with an amplitude of ε = 0.1% and a frequency of 10 Hz. A STARe system differential scanning calorimetric (DSC) instrument (Mettler Toledo, Switzerland) was utilized to test the DSC data of the PU-based composite, which scanned from −100 to 100 °C at a rate of 10 °C/min under N_2_ atmosphere. It was found in Figure 10c,d that the glass transition temperature (T_g_) of the PU-based composite measured using DMA and DSC was −9.3 and −41.6°C, respectively.

DMA was also utilized frequently to survey the viscoelastic behavior of the fabricated PU-based composite to obtain and substitute the viscoelastic parameters into the model. The composite was first cooled to −80 °C with a rate of 3 °C/min, held for 10 min, and then scanned with a frequency range from 1 to 100 Hz, and subsequently, heated to −60 °C at the same rate and held for 5 min. After that, the composite was frequency scanned with different temperature steps in turn. Finally, the dynamic mechanical results at 15 °C including storage modulus (E′) and loss factor (tanδ) related to frequency were obtained with data processing of the DMA software, which is displayed in Figure 11. It was apparent that the magnitude of E′ for the composite increased with increasing frequency, showing that the stiffness of the composite improved with the increase in frequency. In addition, the tanδ curve first increased and then decreased with the increasing frequency, owing to the different responsive frequency range of the composite when it was subjected to external vibration.

#### 3.2.2. SAC Measurements and Validity of the Model

The SAC could be acquired using acoustic pulse tube device measurements, and Figure 12 exhibits the structural diagram of the acoustic pulse tube. As shown in Figure 12, the acoustic pulse tube was placed vertically, and the underwater transducer was at the bottom of the tube. After undergoing the compression molding process, the PU-based composite with a cylindrical shape (diameter of 120 mm and thickness of 50 mm, as shown in Figure 13b) was immersed in water for 24 h. Then, the composite was placed at the top of the acoustic pulse tube to test. The frequency range was from 3 to 8 kHz with water temperature of 15 °C and humidity of 52% RH.

The SAC of the PU-based composite containing simulated and experimental results are shown in Figure 13a, and the relative error of the two sets is listed in Figure 13c and Table 2. As displayed in Table 2, the average experimental value of SAC was 0.85888, whereas the average simulated value of SAC was 0.89589, and the average relative error was 4.49%. It indicates that the simulated results acquired by finite element analysis and the numerical results obtained by the sound absorption experiment were in good agreement, thereby offering a strong validation for the model utilized in this work. In other words, it is feasible to investigate the sound absorption performance of the proposed model using finite element analysis.

## 4. Conclusions

In this work, the underwater sound absorption performance of a PU-based composite was investigated via finite element solutions in the frequency range of 0.5 to 10 kHz under different dynamic mechanical parameters. The underwater structure model of the composite was first established, and the validity and feasibility of the model was verified using acoustic pulse tube tests of the prepared PU-based composite. It was found that the sound absorption performance of the composite displayed a close relationship with the dynamic mechanical parameters. In detail, the internal friction of the molecular chain segments was the main reason for the outstanding sound absorption performance of the PU-based composite with relatively low E′; in contrast, the resonance produced by the enhanced structure stiffness was the major contributor to the consumption of sound energy at high frequencies. Furthermore, the increase in tanδ enhanced the energy dissipation capacity of the PU-based composite, and more sound wave energy was converted to internal energy and dissipated, leading to improvement in the sound absorption proprieties. The demonstrated sound absorption mechanism of the composite at different dynamic mechanical parameters and frequency offers beneficial guidance for underwater sound absorption applications in engineering, especially for designing and synthesizing PU-based sound absorption layers.

## Figures and Tables

**Figure 1 ijms-23-14760-f001:**
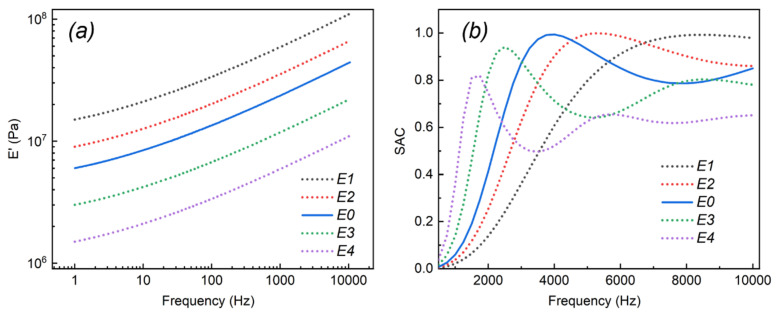
(**a**) E′ settings with different gradients. (**b**) The SAC results of the PU-based composite with different E′.

**Figure 2 ijms-23-14760-f002:**
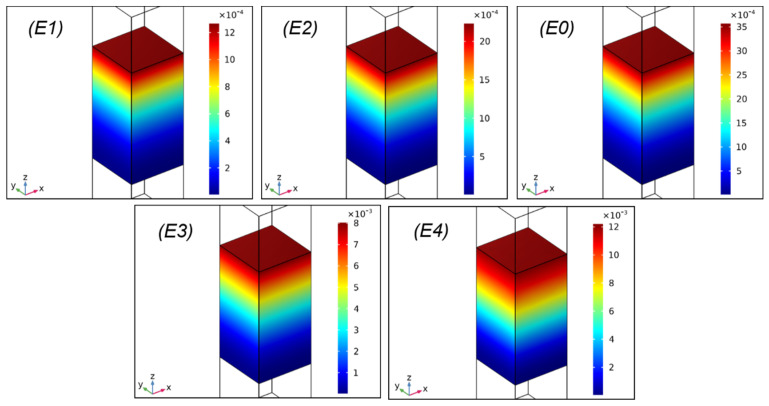
Total power dissipation density (mW/m^3^) of the PU-based composite with different E′ at 1.5 kHz.

**Figure 3 ijms-23-14760-f003:**
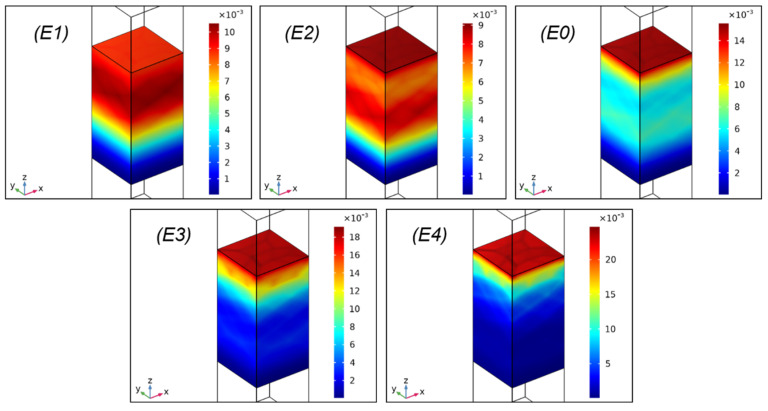
Total power dissipation density (mW/m^3^) of the PU-based composite with different E′ at 10 kHz.

**Figure 4 ijms-23-14760-f004:**
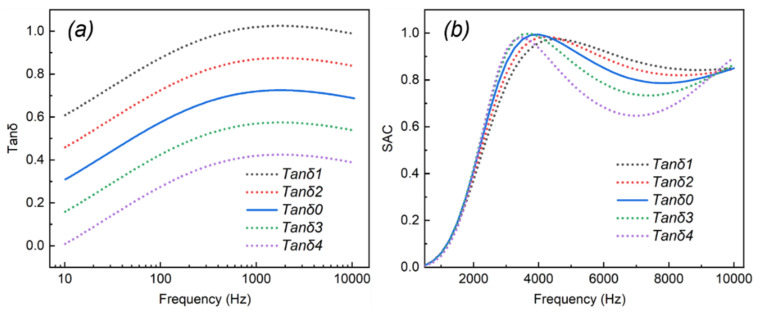
(**a**) Different tanδ settings with gradient. (**b**) The SAC results of the PU-based composite with different tanδ.

**Figure 5 ijms-23-14760-f005:**
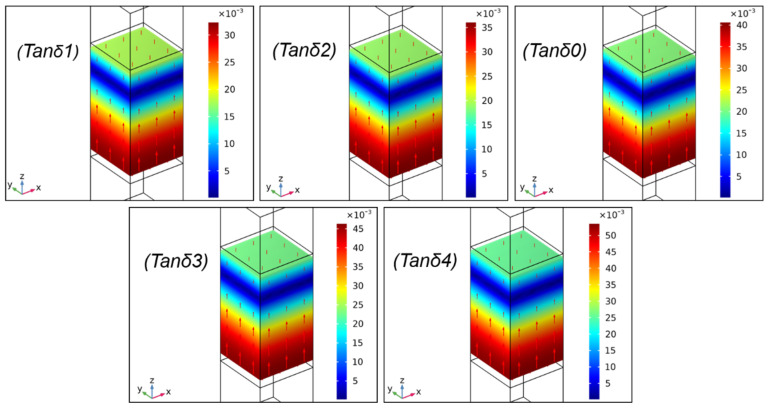
Total displacement (nm) of the composite with different tanδ at 4 kHz.

**Figure 6 ijms-23-14760-f006:**
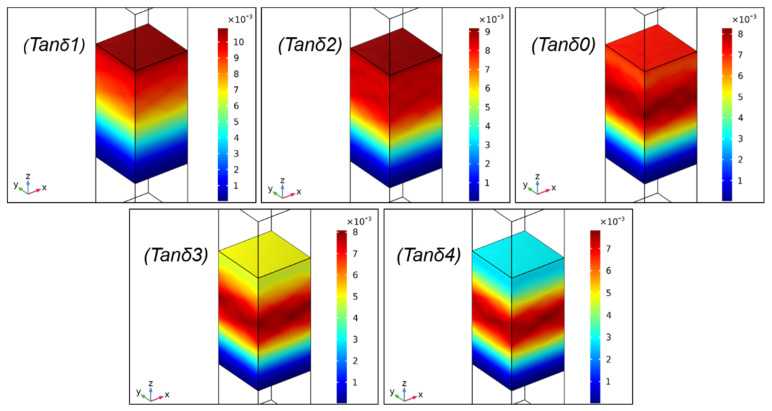
Total power dissipation density (mW/m^3^) of the composite with different tanδ at 7 kHz.

**Figure 7 ijms-23-14760-f007:**
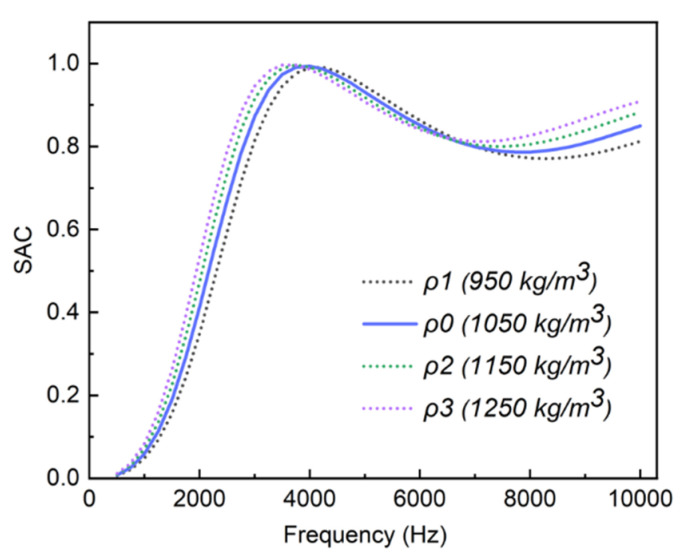
The SAC results of the PU-based composite with different ρ.

**Figure 8 ijms-23-14760-f008:**
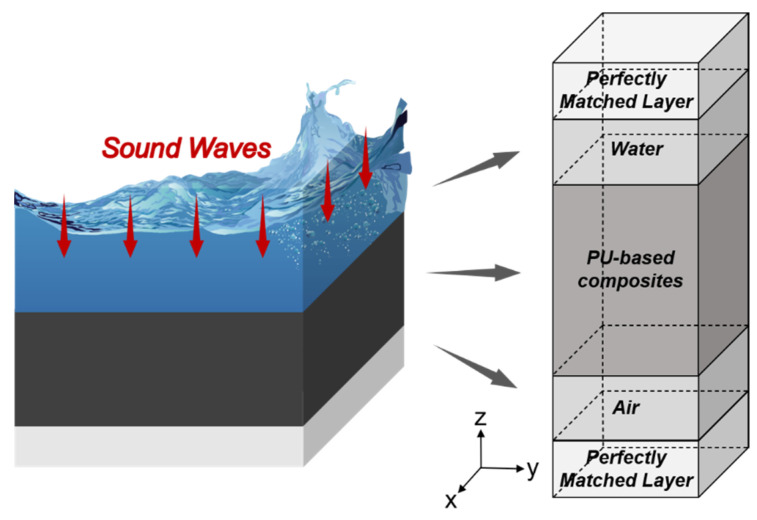
Schematic of the underwater sound absorption layer.

**Figure 9 ijms-23-14760-f009:**
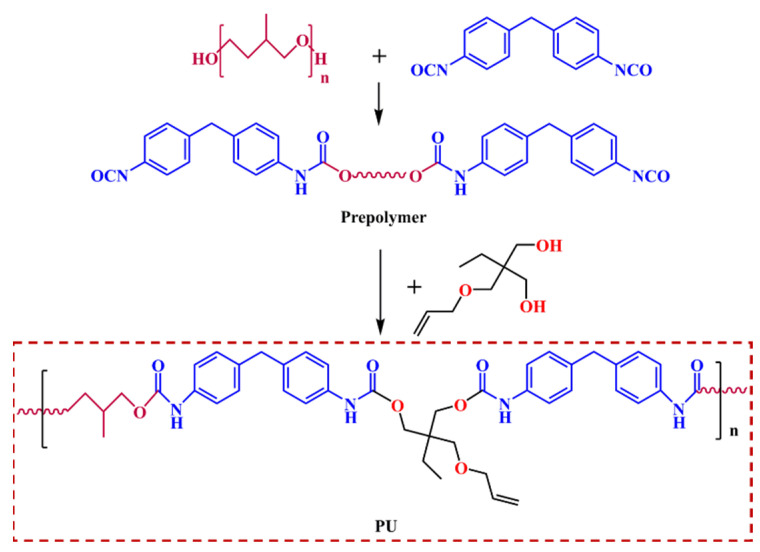
The synthetic route of the PU-based composite.

**Figure 10 ijms-23-14760-f010:**
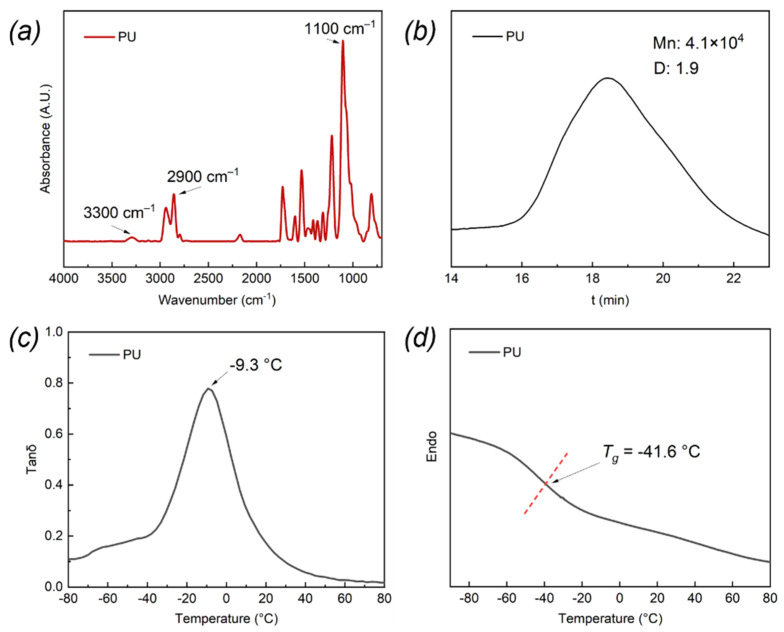
(**a**) FTIR spectrum, (**b**) GPC, (**c**) DMA, and (**d**) DSC curves of the PU-based composite.

**Figure 11 ijms-23-14760-f011:**
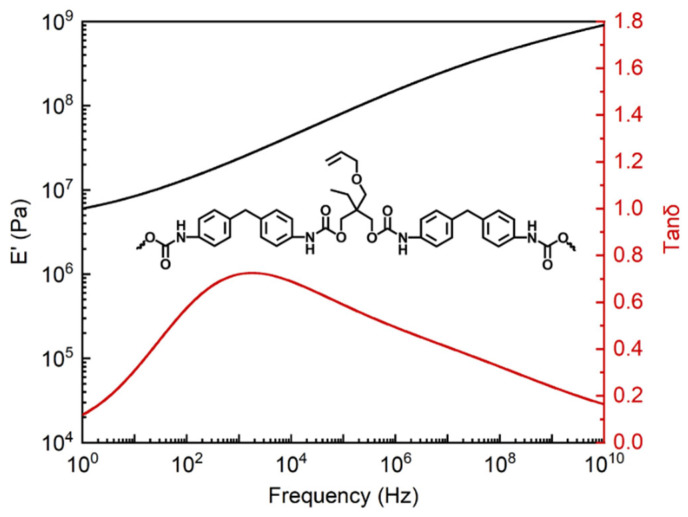
The E′ and tanδ of the PU-based composite.

**Figure 12 ijms-23-14760-f012:**
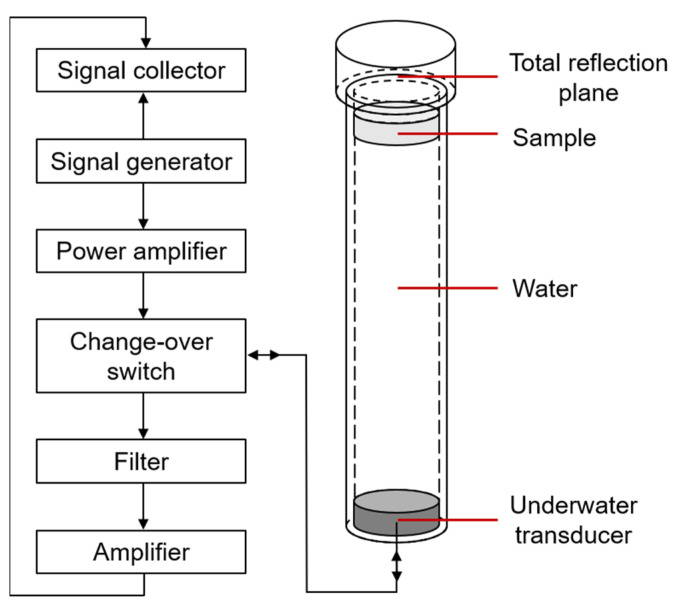
Schematic of the acoustic pulse tube device.

**Figure 13 ijms-23-14760-f013:**
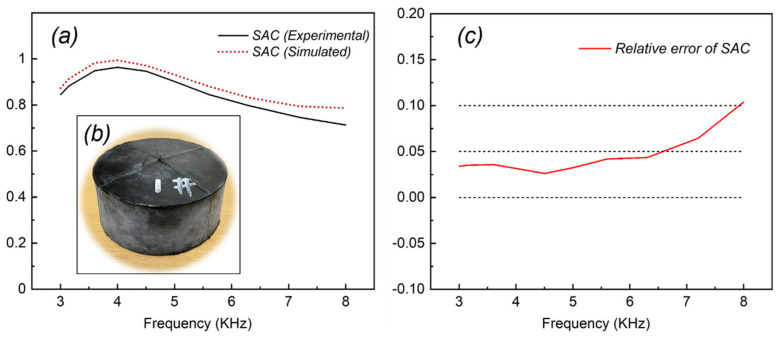
(**a**) Comparison of simulated and experimental sound absorption results; (**b**) the compression-molded PU-based composite; (**c**) the relative error curve of SAC results.

**Table 1 ijms-23-14760-t001:** Material and geometric parameters of the model.

Parameter	Symbol	Value	Unit
Thickness of the composite	*h_d_*	50	mm
Thickness of water layer	*h_w_*	25	mm
Thickness of air layer	*h_a_*	25	mm
Density * of the composite	*ρ*	1050	kg/m^3^
Poisson’s ratio of the composite	*ʋ*	0.49	1
Dynamic modulus * of the composite	*E*	*E*′(*f*_1_)+*i* × *E*″(*f*_2_)	Pa
Density of water	*ρ_w_*	1000	kg/m^3^
Density of air	*ρ_a_*	1.21	kg/m^3^
Sound speed of water	*c_w_*	1500	m/s
Sound speed of air	*c_a_*	343	m/s

* The data were obtained from the prepared composite.

**Table 2 ijms-23-14760-t002:** The relative error data of SAC results.

Frequency (kHz)	SAC (Experimental)	SAC (Simulated)	Relative Error of SAC (%)
3	0.844 ± 0.006	0.873 ± 0.014	3.39
3.15	0.882 ± 0.012	0.913 ± 0.009	3.51
3.6	0.948 ± 0.008	0.982 ± 0.014	3.58
4	0.963 ± 0.014	0.993 ± 0.020	3.15
4.5	0.946 ± 0.023	0.970 ± 0.012	2.61
5	0.901 ± 0.002	0.930 ± 0.015	3.24
5.6	0.845 ± 0.021	0.880 ± 0.003	4.19
6.3	0.797 ± 0.005	0.832 ± 0.006	4.35
7.2	0.745 ± 0.009	0.793 ± 0.002	6.46
8	0.712 ± 0.015	0.786 ± 0.007	10.38
Average value	0.85888	0.89589	4.49

## Data Availability

The data that support the findings of this study are available on request from the corresponding author.

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
