# Peer review of "Finite Element Solution for Dynamic Mechanical Parameter Influence on Underwater Sound Absorption of Polyurethane-Based Composite"

_ijms, 2022, doi:10.3390/ijms232314760_

Round 1
Reviewer 1 Report
This work highlights the underwater sound absorption performance of a PU-based composite. It could provide an excellent reference for sound absorption applications. However, the authors should address some points and suggestions before the work is accepted.
1. The theories in Section 2 lack references.
2. The PU-based composite needs structure and morphological characterization to confirm the successful synthesis.
3. Please put more recent references.
4. Improve the conclusions, which reflect the findings from this study.
Author Response
Reviewer 1
This work highlights the underwater sound absorption performance of a PU-based composite. It could provide an excellent reference for sound absorption applications. However, the authors should address some points and suggestions before the work is accepted.
1.The theories in Section 2 lack references.
Reply: Thanks for the kind suggestion of reviewer #1. The related references have been cited in the revised manuscript. They are as followed [1-6]:
[1] Shi, K.; Jin, G.; Ye, T.; Zhang, Y.; Chen, M.; Xue, Y., Underwater sound absorption characteristics of metamaterials with steel plate backing. Applied Acoustics 2019, 153, 147-156.
[2] Shi, K.; Jin, G.; Liu, R.; Ye, T.; Xue, Y., Underwater sound absorption performance of acoustic metamaterials with multilayered locally resonant scatterers. Results in Physics 2019, 12, 132-142.
[3] Meng, H.; Wen, J.; Zhao, H.; Wen, X., Optimization of locally resonant acoustic metamaterials on underwater sound absorption characteristics. Journal of Sound and Vibration 2012, 331, (20), 4406-4416.
[4] Huang, L.; Xiao, Y.; Wen, J.; Zhang, H.; Wen, X., Optimization of decoupling performance of underwater acoustic coating with cavities via equivalent fluid model. Journal of Sound and Vibration 2018, 426, 244-257.
[5] Ye, C.; Liu, X.; Xin, F.; Lu, T. J., Influence of hole shape on sound absorption of underwater anechoic layers. Journal of Sound and Vibration 2018, 426, 54-74.
[6] Zhao, H.; Wen, J.; Yang, H.; Lv, L.; Wen, X., Backing effects on the underwater acoustic absorption of a viscoelastic slab with locally resonant scatterers. Applied Acoustics 2014, 76, 48-51.
2.The PU-based composite needs structure and morphological characterization to confirm the successful synthesis.
Reply: The characterizations containing FTIR, GPC, DMA and DSC of the PU-based composite have been added to confirm the successful synthesis in the revised manuscript.
Fig. R1. (a) FTIR spectrum, (b) GPC, (c) DMA and (d) DSC curves of the PU-based composite.
Fourier transform infrared spectra (FTIR) spectra of the PU-based composite was tested in attenuated total reflection mode on an FTIR spectrophotometer with a resolution of 4 cm-1 (Tensor 27, Bruker, Germany). Fig. R1a showed the appearance of the absorption peak of the imino group (N-H) (3300 cm-1), the hydrocarbon group (2900 cm-1) and the ether group (-O-) (1100 cm-1). Additionally, the absorption peak of isocyanate group (-NCO) (2300 cm-1) disappeared. Combined with the result of GPC curve in Fig. R1b, it was clearly that the PU-based composite successfully synthesized. Dynamic mechanical analyzer (DMA) (Q800, TA Instruments, USA) was used to measure the dynamic mechanical properties of the PU-based composite in the range of -100 °C-100 °C at a rate of 3 °C/min with an amplitude of ε = 0.1% and a frequency of 10 Hz. A STARe system differential scanning calorimetric (DSC) instrument (Mettler-Toledo, Switzerland) was utilized to test the DSC data of the PU-based composite, which scanned from -100 °C to 100 °C at a rate of 10 °C/min under N2 atmosphere. It was found in Fig. R1c and R1d that the glass transition temperature (Tg) of the PU-based composite measured by DMA and DSC was -9.3°C and -41.6°C, respectively.
3.Please put more recent references.
Reply: More recent references have been cited in the revised manuscript.
4.Improve the conclusions, which reflect the findings from this study.
Reply: Thanks for the beneficial comments of reviewer #1. The conclusions have been improved in the revised manuscript.

Reviewer 2 Report
The novelity of the work is not well highlighted. There are articles where similar studies have been carried out like DOI: 10.1002/app.47165 The authors should state the novelity based the material or the model used.
2. What type of polyurethane based composite has been considered. This should be highlighted in the introduction section
3. Is the PU-based composite a novel material? How did the authors conclude the chemical formula of the composite?
4. No material characterization is included in the manuscript. The authors should show FTIR and the material characterization of this composite.
Author Response
The novelity of the work is not well highlighted. There are articles where similar studies have been carried out like DOI: 10.1002/app.47165 The authors should state the novelity based the material or the model used.
Reply: Thanks for the professional comments of reviewer #2. Our work synthesized a new type of mixed polyurethane, which provided crosslinking points for reactions with divers vulcanizing agents and could be vulcanized with a variety of molds. And that, the simulated and experimental SAC results of the PU-based composite displayed a very small average relative error of 4.49%. Additionally, the dynamic mechanical parameters used in our work were hypothetic parameters with different gradients, based on the data of prepared composite. The parameters are adjustable and thus possesses great flexibility, which could offer better guidance to develop the fabrication and engineering applications of PU-based composite with outstanding underwater sound absorption performance. In contrast, the polyurethanes used in the paper (DOI: 10.1002/app.47165) were purchased commercial products and the dynamic mechanical parameters of the samples were fixed value, which could not be flexible adjustment under different application requirements.
According to the important suggestion of reviewer #2, the novelty of the PU-based composite prepared in this work has been added in the introduction section of the revised manuscript.
1.What type of polyurethane based composite has been considered. This should be highlighted in the introduction section
Reply: The type of the PU-based composite was mixed polyurethane, which has been supplemented in the introduction section of the revised manuscript.
- Is the PU-based composite a novel material? How did the authors conclude the chemical formula of the composite?
Reply: Yes. The synthetic route of the PU-based composite has been added in the revised manuscript.
Fig. R1. The synthetic route of the PU-based composite.
- No material characterization is included in the manuscript. The authors should show FTIR and the material characterization of this composite.
Reply: Thanks for the beneficial comments of reviewer #2. The characterizations containing FTIR, GPC, DMA and DSC of the PU-based composite have been added to confirm the successful synthesis in the revised manuscript.
Fig. R1. (a) FTIR spectrum, (b) GPC, (c) DMA and (d) DSC curves of the PU-based composite.
Fourier transform infrared spectra (FTIR) spectra of the PU-based composite was tested in attenuated total reflection mode on an FTIR spectrophotometer with a resolution of 4 cm-1 (Tensor 27, Bruker, Germany). Fig. R1a showed the appearance of the absorption peak of the imino group (N-H) (3300 cm-1), the hydrocarbon group (2900 cm-1) and the ether group (-O-) (1100 cm-1). Additionally, the absorption peak of isocyanate group (-NCO) (2300 cm-1) disappeared. Combined with the result of GPC curve in Fig. R1b, it was clearly that the PU-based composite successfully synthesized. Dynamic mechanical analyzer (DMA) (Q800, TA Instruments, USA) was used to measure the dynamic mechanical properties of the PU-based composite in the range of -100 °C-100 °C at a rate of 3 °C/min with an amplitude of ε = 0.1% and a frequency of 10 Hz. A STARe system differential scanning calorimetric (DSC) instrument (Mettler-Toledo, Switzerland) was utilized to test the DSC data of the PU-based composite, which scanned from -100 °C to 100 °C at a rate of 10 °C/min under N2 atmosphere. It was found in Fig. R1c and R1d that the glass transition temperature (Tg) of the PU-based composite measured by DMA and DSC was -9.3°C and -41.6°C, respectively.
